# VTG-Reasoner: Long Video Temporal Grounding via Reinforcement Fine-Tuning

## Abstract

Video temporal grounding aims to identify the start and end timestamps of a target event in videos with varying duration. Most of existing methods are trained mainly on short videos through supervised fine-tuning. It is challenging for them to handle long videos, which show diverse data distributions, and thus require to perform reasoning with semantic cues. To conquer this challenge, we propose VTG-Reasoner, a reinforcement fine-tuning framework to enhance the model's reasoning ability for long video temporal grounding. Instead of directly supervising model outputs, VTG-Reasoner explores multiple temporal grounding predictions based on video contexts through an explicit reasoning process. These exploration predictions are then evaluated by our proposed IoU and Intersection Compactness reward to optimize the model. To further enhance the reasoning performance, we adopt relative frame number to replace absolute timestamps, providing a unified temporal representation for videos with varying duration. Quantitative results demonstrate that VTG-Reasoner achieves superior performance on four long video temporal grounding benchmarks in a zero-shot manner, outperforming SFT-based models trained with $20\times$ amount of data.

## 1 Introduction

Video temporal grounding is a fundamental task in video understanding. Given a textual event query, the goal is to predict the start and end timestamps of the corresponding event within the video. Unlike semantic question answering, temporal grounding requires fine-grained modeling of both visual content and temporal information. Accurate temporal grounding is essential for understanding long videos precisely (Liu et al., 2025a) and holds great potential in real-world applications such as autonomous driving and embodied AI.

Multi-modal large language models (MLLM) have demonstrated strong capabilities across a wide range of vision-language tasks (Liu et al., 2023; Achiam et al., 2023; Bai et al., 2025). Recently, many methods have incorporated MLLM for video temporal grounding (Ren et al., 2024; Zeng et al., 2024; Guo et al., 2025b). These methods typically adopt an end-to-end training paradigm, *i.e.*, given multiple frames of visual input and a textual query, the model directly outputs the start and end timestamps of the target event. Currently, these methods are mostly trained on short videos and are also tested on short-video benchmarks.

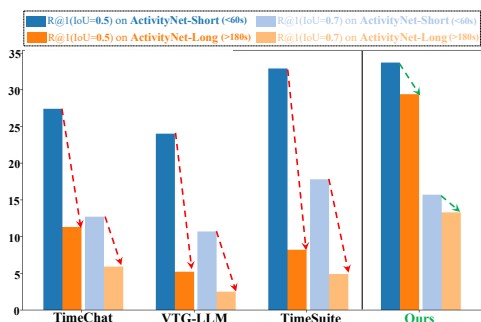

Figure 1: The left side shows decreased performance of existing video temporal grounding methods on long videos. The right side shows that our method demonstrates robust performance across varying video lengths.

Although MLLM-based approaches have achieved promising results on short videos, they struggle to achieve satisfactory grounding performance on long videos. As shown in Figure 1, based on the ActivityNet Captions dataset (Caba Heilbron et al., 2015), we construct the ActivityNet-Short subset by selecting videos shorter than 60 seconds and the ActivityNet-Long subset by selecting videos longer than 180 seconds. We evaluate three repre-

sentative methods (Ren et al., 2024; Zeng et al., 2024; Guo et al., 2025b) on two subsets of varying lengths. The results show a clear performance drop as the video length increases. The above phenomenon indicates that existing methods struggle to handle long video temporal grounding when trained mainly on short videos.

The above challenge can be attributed to the fact that, long videos contain richer visual cues and more dispersed event distributions, which require the model to perform temporal reasoning conditioned on visual context. Existing temporal grounding models lack such reasoning capability largely due to two aspects: **training paradigm** and **temporal representation**. (1) Existing methods widely adopt supervised fine-tuning to optimize the model, which is prone to memorizing the training data distribution and suffers from out-of-distribution data. This makes models lack reasoning capability and struggle to process more complex and longer videos. (2) Absolute timestamps make it difficult to provide a unified temporal representation for videos with different durations. Predictions using absolute timestamps are more likely to produce hallucinations on long videos.

To conquer this challenge, we propose VTG-Reasoner, a reinforcement fine-tuning (RFT) framework to enhance the model's reasoning ability for long video temporal grounding. Unlike SFT that directly supervises model output, VTG-Reasoner allows model to perform extensive exploration and trial-and-error to generate a group of predictions through an explicit reasoning process. These multiple predictions enable the model to explore different temporal grounding results based on visual context and textual query on long videos, instead of simply memorizing the training data distribution. To guide this process, we design two reward functions, *i.e.*, an IoU reward to encourage accurate alignment with ground truth intervals, and an Intersection Compactness reward to avoid greedy predictions and promote concise localization.

To further enhance the reasoning performance, we use relative frame order to replace absolute timestamps. This modification transforms temporal grounding into a problem of reasoning over frame positions rather than predicting precise timestamps. This ensures a unified representation across videos of diverse lengths. It thus simplifies the temporal grounding process and makes it more robust to variations in video length. Those components improve the temporal reasoning capability of VTG-Reasoner by encouraging it to jointly leverage visual contexts and relative frame order, rather than relying on the memorization of patterns between visual features and absolute timestamps. With better reasoning ability, our approach exhibits superior generalization capability to handle long videos.

To better simulate the real scenarios, where the annotations in long videos are hard to acquire, we further propose a more challenging setup named **Short-to-Long**. Short-to-Long trains the model on a limited number of short videos and performs evaluation on long videos with more complex and diverse semantics. Specifically, the duration of training videos is limited to within 60 seconds, whereas the average duration of four evaluation benchmarks ranges from 150 to 368 seconds. This new setting is designed to assess the model's generalization ability under limited training data, as well as the effectiveness in handling more difficult temporal grounding cases.

Under the Short-to-Long setup, VTG-Reasoner outperforms previous methods on four long video temporal grounding benchmarks in a zero-shot manner. VTG-Reasoner also exhibits higher data efficiency. Using 16K training samples, it outperforms TimeSuite (Zeng et al., 2024) trained with 375K samples and HawkEye (Wang et al., 2024) trained with 715K samples. To the best of our knowledge, this is an original effort on the Short-to-Long setup for video temporal grounding. The strong performance enhancement clarifies the effectiveness of our VTG-Reasoner. Our dataset and model will be released.

## 2 RELATED WORK

### 2.1 TEMPORAL GROUNDING BASED ON MLLM

Traditional methods rank the candidate proposals (Yuan et al., 2019; Zhang et al., 2020) or regress the start and end boundaries (Zeng et al., 2020; Lin et al., 2023). Temporal grounding methods based on MLLM usually adopt an end-to-end paradigm. VtimeLLM (Huang et al., 2024) proposes a three-stage instruction tuning framework based on a temporal-aware dialogue dataset. TimeChat (Ren et al., 2024) combines temporal information and video frames through Q-Former (Li et al., 2023). Momentor (Qian et al., 2024) uses a temporal perception module to explicitly model time and inject

time embedding into training. HawkEye (Wang et al., 2024) designs a coarse-grained representation of video clips and proposes a recursive grounding technique. VTG-LLM (Guo et al., 2025b) incorporates absolute-time tokens to manage timestamp knowledge. TimeSuite (Zeng et al., 2024) and VideoChat-Flash (Li et al., 2024b) equip models with both temporal grounding and semantic understanding capabilities through efficient compression mechanisms and multi-task datasets.

These methods rely on supervised fine-tuning for training and often require complex temporal modeling mechanisms and additional learnable tokens, resulting in weak generalization and data efficiency. To address above limitations, we enhance the temporal reasoning capability of model at both the training paradigm and temporal representation levels.

## 2.2 Reinforcement Fine-tuning in Vision Tasks

Reasoning models such as openAI o1 (Jaech et al., 2024) and DeepSeek-R1-Zero (Guo et al., 2025a) have demonstrated that reinforcement fine-tuning (RFT) can effectively improve the reasoning ability of LLM. Subsequent work extends it to multi-modal tasks (Huang et al., 2025; Feng et al., 2025). Based on the Group Relative Policy Optimization (GRPO) algorithm (Shao et al., 2024), MLLM can solve more complex problems after post-training with rule-based reward. The current work focuses mainly on two types of tasks. The first type is visual reasoning tasks such as mathematical reasoning problems (Meng et al., 2025; Peng et al., 2025). The second type is visual perception tasks. Visual-RFT (Liu et al., 2025b) and Reason-RFT (Tan et al., 2025) explore the application of RFT in object detection and geometric understanding tasks. There are also studies that have initially tried the application of RFT in video temporal grounding (Wang et al., 2025; Li et al., 2025). Their works remain focused on the short-video domain, whereas our study targets long video temporal grounding and further explores reward function design, temporal representation and evaluation settings.

## 3 Proposed Method

### 3.1 Preliminary of Reinforcement Fine-tuning

Supervised fine-tuning (SFT) and reinforcement fine-tuning (RFT) are widely used post-training paradigms for MLLM. Compared to SFT that tends to memorize the training data distribution, RFT learns through exploration and trial-and-error. RFT thus presents better data efficiency and generalization performance.

Group Relative Policy Optimization (GRPO) is a commonly used RFT algorithm (Shao et al., 2024). Other RFT algorithms such as PPO (Schulman et al., 2017) require an additional value model to evaluate the model's answers. GRPO uses group-averaged baselines for advantage estimation, saving considerable computing resources. Specifically, let $\pi_\theta$ be the policy model and let $R$ be the task-specific reward function. For an input $q$, $\pi_\theta$ first samples a group of $G$ responses $o = \{o_1, o_2, \ldots, o_G\}$, then assigns corresponding rewards $\{r_1, r_2, \ldots, r_G\}$ to responses according to the reward function $R$. The GRPO algorithm regularizes this set of rewards to determine the quality of the response, *i.e.*,

$$A_i = \left(r_i - \text{mean}\left(\{r_i\}_{i=1}^G\right)\right) / \text{std}\left(\{r_i\}_{i=1}^G\right), \tag{1}$$

where $A_i$ represents the relative advantage of the $i$-th response. $A_i$ will be used for token-level optimization of the policy model. The final optimization objective is as follows:

$$\mathcal{J}(\theta) = \frac{1}{G} \sum_{i=1}^G \frac{1}{|o_i|} \sum_{t=1}^{|o_i|} \left[ \min\left(r_t(\theta)A_{i,t}, \text{ clip}(r_t(\theta), 1-\epsilon, 1+\epsilon) A_{i,t}\right) - \beta D_{\text{KL}}(\pi_\theta \| \pi_{\text{ref}}) \right], \tag{2}$$

$$r_t(\theta) = \frac{\pi_\theta\left(o_{i,t} \mid q, o_{i,<t}\right)}{\pi_{\theta_{\text{old}}}\left(o_{i,t} \mid q, o_{i,<t}\right)}, \tag{3}$$

where $\beta$ and $\epsilon$ are coefficients controlling the KL divergence and the clipping range, respectively, while $\pi_{\theta_{\text{old}}}$ and $\pi_{ref}$ are the old policy model and the reference policy model, respectively. The KL divergence is introduced to prevent the policy model from being updated excessively.

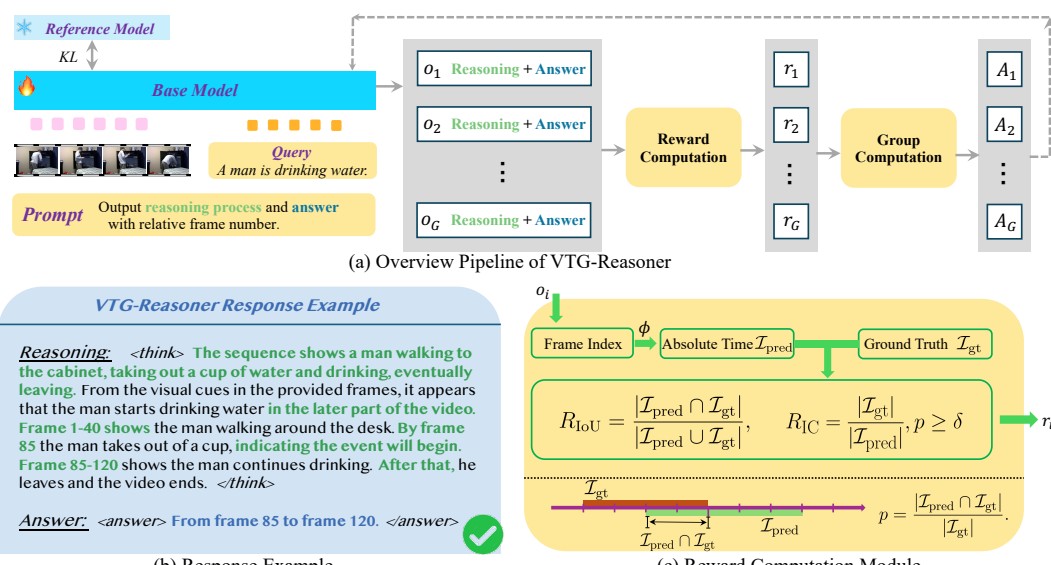

(a) Overview Pipeline of VTG-Reasoner

(b) Response Example

(c) Reward Computation Module

Figure 2: Overview of VTG-Reasoner. (a) shows the pipeline of VTG-Reasoner. (b) presents a response example containing reasoning process and final answer. The green texts highlight the correct reasoning path of VTG-Reasoner. (c) illustrates the reward computation module. It first converts the relative frame order from the answer into absolute timestamps, then calculates the reward with the ground truth based on two reward functions.

## 3.2 OVERVIEW OF VTG-REASONER

Figure 2(a) provides an overview of VTG-Reasoner. It is built upon the GRPO algorithm, aiming to enhance the model's reasoning ability for long video temporal grounding. Given a video of length $L$, it uniformly samples $N$ frames. The event is specified by a query $Q$, accompanied by a prompt $P$. The sampled frames are encoded into visual tokens using a vision encoder, while the query and prompt are tokenized into text tokens. These tokens are then jointly fed into the LLM backbone. Using temperature sampling, the model generates a set of $G$ responses $\{o_1, o_2, \ldots, o_G\}$. This suggests that the model conducts $G$ rounds of exploration over the video and query, making it more suitable for long video scenarios with complex and diverse semantics.

As shown in Figure 2(b), each response $o_i$ includes an explicit chain-of-thought within `<think>` tags and predicted frame indices within `<answer>` tags. Next, a reward $r_i$ is assigned to each response $o_i$ by our proposed reward functions. As shown in Figure 2(c), in the reward computation module, the relative frame order is first converted to absolute timestamps, which are then compared with the ground truth to compute the reward. Finally, all rewards $\{r_1, r_2, \ldots, r_G\}$ are transformed into a group of advantages $\{A_1, A_2, \ldots, A_G\}$ through group computation, which are subsequently used to update the base model.

The following sections present details of reward design, the computation of relative frame order, as well as the training and evaluation setup.

## 3.3 REWARD DESIGN

To guide the reasoning process in video temporal grounding, we leverage two types of rewards, *i.e.*, IoU reward and Intersection Compactness reward (IC reward). The IoU reward is the primary reward function suitable for temporal grounding. Meanwhile, we observe a phenomenon of greedy predictions during training. We hence introduce the IC reward to encourage more precise and fine-grained grounding results. The final reward function is computed as follows:

$$R = R_{\text{IoU}} + w \cdot R_{\text{IC}}, \tag{4}$$

where $R_{\text{IoU}}$ is the IoU reward, $R_{\text{IC}}$ is the Intersection Compactness reward, $w$ is the weight coefficient.

**IoU Reward.** Both predictions and ground truth are represented as time intervals. The Intersection over Union (IoU) reward is to encourage these two intervals to overlap as much as possible. It can be represented as:

$$R_{\text{IoU}} = \frac{|\mathcal{I}_{\text{pred}} \cap \mathcal{I}_{\text{gt}}|}{|\mathcal{I}_{\text{pred}} \cup \mathcal{I}_{\text{gt}}|}, \tag{5}$$

where $\mathcal{I}_{\text{pred}}$ and $\mathcal{I}_{\text{gt}}$ represent the prediction interval and the ground truth interval, respectively.

**IC Reward.** During training with IoU reward, we observe a greedy prediction phenomenon, where the model tends to predict overly large intervals to cover the ground truth segment. This behavior leads to coarse-grained grounding results and limits the model's ability to handle long videos.

To address this issue, we introduce the IC reward to promote more compact and precise temporal grounding. This reward encourages the model to generate shorter prediction intervals only when its prediction is relatively accurate. Specifically, we first define the coverage proportion $p$ between the predicted and ground truth intervals:

$$p = \frac{|\mathcal{I}_{\text{pred}} \cap \mathcal{I}_{\text{gt}}|}{|\mathcal{I}_{\text{gt}}|}. \tag{6}$$

Let $\delta \in (0, 1)$ be a threshold hyperparameter. If $p \geq \delta$, the prediction is considered successful. In this case, we calculate the ratio between the ground truth length and the predicted length as the reward, which encourages a more compact prediction. In contrast, if $p < \delta$, the prediction is considered as a failure and the IC reward is set to zero. With this reward, the model is driven to make compact predictions under the condition of maintaining correct localization. We define the reward function as follows:

$$R_{\text{IC}} = \begin{cases} \dfrac{|\mathcal{I}_{\text{gt}}|}{|\mathcal{I}_{\text{pred}}|}, & \text{if } p \geq \delta \\ 0, & \text{if } p < \delta \end{cases}. \tag{7}$$

### 3.4 RELATIVE FRAME ORDER

Previous works typically use absolute timestamps (*e.g.*, from 10 s to 20 s) for video temporal grounding. Such representations could limit the model's capability to generalize to long videos, especially when there is a significant difference in video lengths between the training and testing sets. In this paper, we adopt the relative frame order as the temporal representation.

Specifically, for a video of length $L$ seconds, we uniformly sample $N$ frames as visual input to the MLLM. During temporal grounding, the model is required to predict the start and end frame indices of the target event. In this way, the model's output shifts from continuous absolute timestamps (in seconds) to discrete frame indices. We define a mapping function $\phi$ to convert relative frame order to absolute timestamps. Thanks to uniform sampling, $\phi$ is a simple linear transformation. Given a frame index $f$, the corresponding absolute time $t$ can be calculated as:

$$t = \phi(f) = \frac{(f-1)L}{N}, \quad f \in [1, N]. \tag{8}$$

This formulation offers two main advantages. First, it provides a unified temporal representation across videos of diversified lengths, alleviating the domain gap between short and long videos. Second, the use of discrete frame indices defines a smaller exploration space, thereby reducing the learning difficulty of the model. As shown in Figure 3, the model exhibits a vague understanding of absolute time, and may even experience severe hallucinations. However, the model with relative frame order can perceive the relative order of events and identify the corresponding frame indices.

Studies such as Vid2Seq (Yang et al., 2023) and Seq2Time (Deng et al., 2025) have also explored the idea of using relative temporal representations. However, their methods require introducing additional temporal tokens `<0><1>...<9>` and constructing large-scale datasets to learn the representations of these tokens. This is difficult to achieve given the current lack of long videos with temporal annotations. In contrast, our method does not introduce any extra tokens or temporal injection modules. Instead, we express relative temporal concepts purely through natural language and achieve favorable results within the RFT framework.

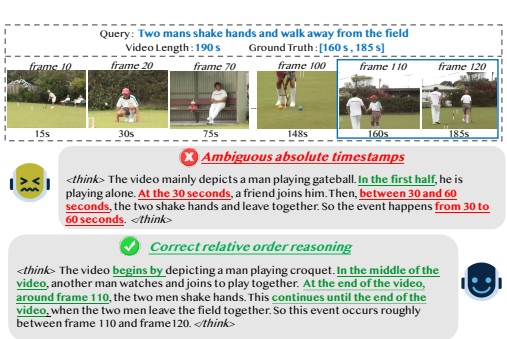

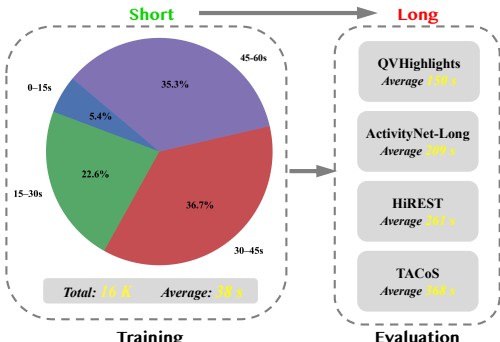

Figure 3: Reasoning process of different temporal representations. Reasoning with absolute timestamps may exhibit hallucinations, while relative frame order makes correct judgments.

Figure 4: Details of Short-to-Long setting. The left side shows the distribution of training data. The right side presents the test datasets and their average lengths.

## 3.5 TRAINING AND EVALUATION SETUP

Existing SFT-based methods adopt a common training and evaluation protocol, where the training employs extensive data from multiple time-sensitive tasks. Moreover, the training set does not enforce strict constraints on video duration and may incorporate a substantial proportion of long-form videos. The testing is conducted on short-form videos (*e.g.*, Charades-STA (Gao et al., 2017) with an average length of 30 seconds).

To better assess the model's generalization under varying video lengths, we propose a more challenging setup called **Short-to-Long**. Compared with the original setup, this setup presents two differences: i) the amount of training data is limited; ii) there is a clear gap between the lengths of training and testing videos. Specifically, the Short-to-Long setting employs a much smaller training set of 16K samples with all videos no longer than 60 seconds, while the average duration of four evaluation benchmarks ranges from 150 to 368 seconds. Figure 4 shows details of our setting. This Short-to-Long setup aims to simulate real-world application scenarios, where temporal grounding of long videos is more meaningful, and annotations for long videos are scarce and difficult to obtain. We also expect that this more challenging setting can provide a comprehensive evaluation of the model's generalization ability and data efficiency.

## 4 EXPERIMENTS

### 4.1 EXPERIMENTS SETTINGS

**Training Setup.** Under the Short-to-Long setting, we first collect videos shorter than 60 seconds from InternVid (Wang et al., 2023) dataset as the source data. Then, inspired by FAST-GRPO (Xiao et al., 2025), we filter the original dataset to obtain samples with moderate difficulty. Specifically, we use the IoU between the base model's predictions and the ground truth as the filtering criterion. Details of the filtering process are provided in the Appendix B. After filtering, we finally get **16K** training samples, with an average video length of **38** seconds. As shown in Figure 4, our training dataset is constrained to the domain of short videos, with no video exceeding 60 seconds in duration.

**Evaluation Setup.** We select four long video temporal grounding benchmarks for evaluation. (1) QVHighlights (Lei et al., 2021) with an average video duration of **150** seconds. (2) ActivityNet-Long with an average video length of **209** seconds. We select videos longer than 180 seconds from the ActivityNet Captions (Caba Heilbron et al., 2015) test set to construct this dataset. (3) HiREST (Zala et al., 2023) with an average video duration of **261** seconds. (4) TACoS (Regneri et al., 2013) with an average video length of **368** seconds. Following privious work, we use Recall@1 at different IoU thresholds as evaluation metrics. For QVHighlights and ActivityNet-Long datasets, we adopt Recall@1 at IoU thresholds of 0.5 and 0.7. For the longer and more challenging HiREST and TACoS datasets, we report Recall@1 at IoU thresholds of 0.3 and 0.5.

Table 1: Performance comparison on four long video temporal grounding benchmarks.

| Method | Training Size | ActivityNet-Long | | QVHighlights | | HiREST | | TACoS | |
|--------|---------------|------------------|------|--------------|------|--------|------|-------|------|
| | | R@0.5 | R@0.7 | R@0.5 | R@0.7 | R@0.3 | R@0.5 | R@0.3 | R@0.5 |
| *General Video Understanding MLLMs* | | | | | | | | | |
| Video-ChatGPT | – | 0.7 | 0.4 | 0.5 | 0.1 | 1.7 | 0.6 | 5.6 | 1.9 |
| VideoLLaMA2 | – | 6.7 | 2.8 | 8.0 | 3.0 | 1.9 | 0.6 | 7.3 | 3.7 |
| *Video Temporal Grounding MLLMs* | | | | | | | | | |
| TimeChat | 125 K | 11.3 | 5.9 | 9.4 | 4.1 | 2.8 | 1.1 | 3.6 | 1.5 |
| VTimeLLM | 170 K | 25.4 | 10.2 | 26.2 | 11.4 | 6.5 | 1.9 | 9.8 | 4.3 |
| HawkEye | 715 K | 22.7 | 9.1 | 14.0 | 4.7 | 6.5 | 1.5 | 9.4 | 4.0 |
| VTG-LLM | 120 K | 5.2 | 2.5 | 6.7 | 2.4 | 2.7 | 1.2 | 6.7 | 2.9 |
| TimeSuite | 349 K | 8.2 | 4.9 | 12.2 | 9.0 | 3.0 | 0.9 | 6.8 | 2.6 |
| VideoChat-Flash | 700 K | 14.2 | 7.9 | 23.9 | 13.9 | 3.1 | 1.7 | 12.2 | 5.3 |
| VTG-Reasoner | **16 K** | **29.4** | **13.3** | **34.3** | **19.0** | **8.7** | **3.8** | **16.1** | **6.9** |

**Implementation Details.** We use Qwen2.5-VL-7B-Instruct (Bai et al., 2025) as the base model. During training, we uniformly sample 128 frames and the resolution is $224 \times 224$. We train the model on $8 \times$ A100 GPUs for 1 epoch. We set $\delta = 0.5$ and the weight $w = 0.5$ for IC reward. During evaluation, we sample 128 frames for long videos and 64 frames for short videos. More implementation details can be found in Appendix C.

## 4.2 COMPARISON METHODS

We categorize the comparison methods into two groups. The first group consists of general video understanding MLLMs, including Video-ChatGPT (Maaz et al., 2023), VideLLaMA2 (Cheng et al., 2024). The second group comprises video temporal grounding MLLMs, including TimeChat (Ren et al., 2024), VTimeLLM (Huang et al., 2024), HawkEye (Wang et al., 2024), Momentor (Qian et al., 2024), VTG-LLM (Guo et al., 2025b), VideoChat-Flash (Li et al., 2024b) and TimeSuite (Zeng et al., 2024). All models are 7B scale. For HawkEye, we set grounding rounds as 1 to align with other models. Since Momentor has not released its checkpoints, we are unable to report its performance for comparison.

Retraining all models under the same benchmark is not feasible. Therefore, we perform evaluation using their publicly released checkpoints under identical testing conditions. Although the training data for each model differs, we emphasize that our training setup is at a disadvantage, e.g., our 16K training dataset is a subset of the 170K samples used by VTimeLLM and the 715K samples used by HawkEye.

## 4.3 RESULTS ON LONG VIDEO TEMPORAL GROUNDING

**Performance and Analysis.** We present the quantitative results in Table 1. Compared with previous methods, VTG-Reasoner achieves superior performance across four long video temporal grounding benchmarks in a zero-shot manner. Note that, those compared temporal grounding models mostly have included the ActivityNet-Long dataset for training. In contrast, our method is evaluated in a zero-shot setting, but still achieves improvements of 4.0 and 3.1 in R@1 at IoU thresholds of 0.5 and 0.7, respectively.

We can draw three conclusions from the experimental results: **i) VTG-Reasoner demonstrates strong generalization ability from short to long videos.** Although our model is trained on a dataset with an average length of 38 seconds, it can outperform other temporal grounding methods on videos that are **10×** longer in duration. **ii) VTG-Reasoner exhibits higher data efficiency.** SFT-based methods require large-scale datasets for training, with VTimeLLM and HawkEye utilizing 170K and 715K samples, respectively. In contrast, VTG-Reasoner achieves superior performance using only a 16K subset derived from their training datasets. **iii) VTG-Reasoner does not compromise general video understanding capability.** Table 2 presents a performance comparison on general VQA benchmarks MVBench (Li et al., 2024a) and VideoMME (Fu et al., 2025). VTG-Reasoner achieves superior performance on two benchmarks, outperforming certain general

Table 2: The performance comparison on general VQA benchmarks.

| Model | MVBench | VideoMME |
|---|---|---|
| | Avg | Avg, w/o subs |
| *General Video Understanding Models* | | |
| VideoLLaMA2 | 54.6 | 46.6 |
| VideoChat2 | 60.4 | 39.5 |
| *Video Temporal Grounding Models* | | |
| TimeChat | 30.2 | 38.5 |
| TimeSuite | 46.3 | 59.9 |
| VTG-Reasoner | **67.8** | **61.1** |

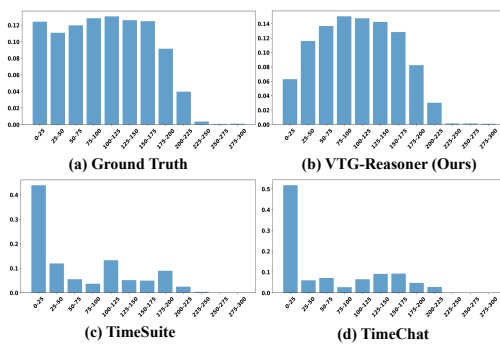

Figure 5: Visualization of prediction intervals distribution.

video understanding models and SFT-based methods. This suggests that the RFT framework does not enhance the model's domain-specific capabilities at the expense of its original performance.

**Visualization.** We further visualize the prediction distributions of three methods on the ActivityNet-Long dataset. Figure 5 shows that VTG-Reasoner produces a more uniform prediction distribution, which aligns more closely with the ground truth distribution. In contrast, TimeSuite and TimeChat tend to concentrate predictions within shorter temporal segments, leading to degraded performance. The visualization results indicate that SFT-based methods tend to memorize the distribution of the training data, whereas VTG-Reasoner demonstrates stronger generalization capability from short to long videos.

### 4.4 RESULTS ON SHORT VIDEO TEMPORAL GROUNDING

To ensure a comprehensive comparison, we also conduct experiments on short video temporal grounding benchmark, where many previous works are specifically optimized. Following previous work, we test on the Charades-STA dataset (Gao et al., 2017), with an average video length of 30 seconds. We report the Recall@1 at IoU thresholds of 0.5 and 0.7 as evaluation metrics. The results are presented in Table 3. We compare our method with TimeChat (Ren et al., 2024), VTimeLLM (Huang et al., 2024), HawkEye (Wang et al., 2024), Momentor (Qian et al., 2024), VTG-LLM (Guo et al., 2025b). The results show that VTG-Reasoner outperforms other methods in terms of R@0.5 and ranks second at R@0.7, suggesting its effectiveness in short video temporal grounding. This indicates that VTG-Reasoner can handle videos of varying lengths, while also demonstrating promising potentials in long video temporal grounding.

### 4.5 ABLATION STUDIES

This section conducts ablation studies on the QVHighlights dataset. Additional ablation studies are presented in the Appendix A, including analyses on the parameters of reward function, the number of test frames, and the model size.

**Comparison between RFT and SFT.** Table 4 shows that, SFT brings only limited improvements over the base model. RFT significantly enhances the model's temporal grounding performance on long videos. This could be because SFT tends to memorize training data, while RFT leverages multiround rollouts and a well-designed reward to encourage exploration, thus presents better reasoning ability.

**Comparison between Relative Frame Number and Absolute Timestamps.** The comparison between line 3 and line 4 in Table 4 shows that, the relative frame order improves R@0.5 and R@0.7 by 5.7 and 1.9, respectively. This indicates that the relative frame order unifies the temporal representation across videos of varying lengths, making it more suitable for long video scenarios.

**Ablation of Reward Design.** While using the IC reward or the IoU reward individually results in comparable performance, their combined utilization demonstrates superior effectiveness, yielding

Table 3: Performance comparison on short video benchmark.

| Method | Charades-STA | |
|---|---|---|
| | R@0.5 | R@0.7 |
| TimeChat | 32.3 | 13.4 |
| VTimeLLM | 27.5 | 11.4 |
| HawkEye | 31.4 | 14.5 |
| Momentor | 26.6 | 11.6 |
| VTG-LLM | 33.8 | **15.7** |
| VTG-Reasoner | **36.6** | 14.8 |

Table 4: Ablation studies of training paradigm, temporal representation and reward design.

| Train | Time | Reward | QVHighlights | |
|---|---|---|---|---|
| | | | R@0.5 | R@0.7 |
| Baseline | – | – | 13.6 | 7.2 |
| SFT | absolute | – | 16.5 | 10.8 |
| RFT | absolute | IoU | 25.5 | 14.9 |
| RFT | relative | IoU | 31.2 | 16.8 |
| RFT | relative | IC | 29.0 | 16.1 |
| RFT | relative | IoU + IC | 34.3 | 19.0 |

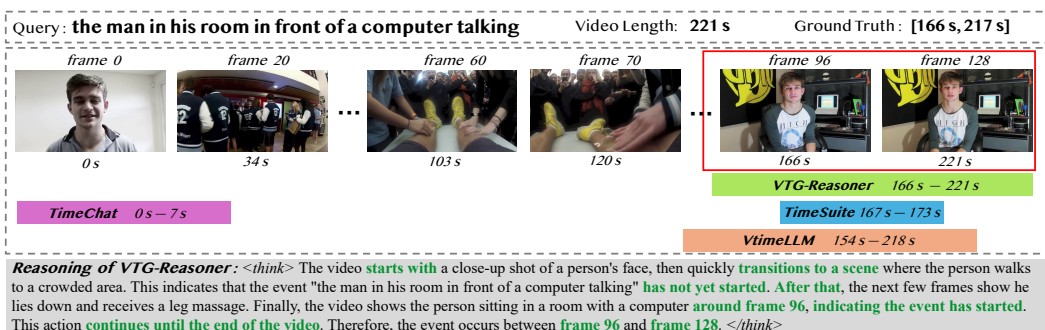

Figure 6: Illustration of the reasoning processing.

the best overall results. The comparison between line 4 and line 6 in Table 4 shows that, the introduction of IC reward further improves performance. We also observe that after introducing the IC reward, the average length of the predicted intervals decreases from 56 seconds to 49 seconds. This indicates that the IC reward effectively mitigates greedy predictions and facilitates more precise localization results.

**Impact of Frame Number.** Table 5 shows that the performance improves as the number of sampled frames increases during training. In short videos, visual information tends to be redundant, whereas long videos contain richer visual cues and more complicated events. Sampling more frames helps the model to better perceive the temporal order of events. However, excessive visual inputs will increase computational overhead. We choose to sample 128 frames during training.

Table 5: Impact of different frame number during training.

| Frame number | QVHighlights | |
|---|---|---|
| | R@0.5 | R@0.7 |
| 32 | 20.5 | 10.4 |
| 96 | 33.3 | 18.8 |
| 128 | 34.3 | 19.0 |

**Case Study.** We provide a case for a qualitative comparison with other methods. As shown in Figure 6, VTG-Reasoner first makes a relatively accurate judgment of the order of events and performs logical reasoning over the video context and events, then produces more accurate prediction than other methods. In the Appendix E, we provide more case studies on three additional test sets, along with a failure case and its in-depth analysis.

## 5 CONCLUSION

In this paper, we propose VTG-Reasoner, a reinforcement fine-tuning framework to enhance the model's reasoning ability for long video temporal grounding. Experimental results demonstrate that VTG-Reasoner exhibits superior generalization and high data efficiency. This paper highlights the potential of reinforcement fine-tuning for advancing long video understanding. We provide further discussion of limitations and future directions in the Appendix D.

ETHICS STATEMENT

Our method is dedicated to improving the model's performance on long video temporal grounding, but we discourage the use of VTG-Reasoner model for encoding or retrieving sensitive content. Furthermore, we strongly oppose applying this model to the processing of violent or harmful videos.

REPRODUCIBILITY STATEMENT

The training and evaluation datasets mentioned in this paper are publicly available. We provide the training data in a JSON file in the supplementary materials, which includes video YouTube IDs, questions, answers, and other relevant information. All video IDs can be accessed and downloaded from the internet. The training and testing code will be made publicly available as soon as it has undergone internal inspection and verification.

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

# A  ADDITIONAL ABLATION RESULTS

## A.1  PARAMETERS OF REWARD DESIGN

The reward function of VTG-Reasoner is as follows:

$$R = R_{\text{IoU}} + w \cdot R_{\text{IC}}, \tag{9}$$

$$R_{\text{IC}} = \begin{cases} \dfrac{|\mathcal{I}_{\text{gt}}|}{|\mathcal{I}_{\text{pred}}|}, & \text{if } p \geq \delta \\ 0, & \text{if } p < \delta \end{cases}, \quad p = \frac{|\mathcal{I}_{\text{pred}} \cap \mathcal{I}_{\text{gt}}|}{|\mathcal{I}_{\text{gt}}|}. \tag{10}$$

As presented in Table 6, we conduct ablation studies on $\delta$ and $w$. Specifically, $w$ is the coefficient balancing the two reward functions, while $\delta$ is the threshold parameter that determines whether the IC reward is set to 0. Both parameters range from 0 to 1.

We present two key findings based on the experimental results: (1) A small $\delta$ leads to a significant performance drop. We attribute this phenomenon to reward hacking. In this case, the model can obtain higher rewards simply by predicting shorter intervals, thus diminishing the impact of the IoU-based reward signal. (2) The performance remains robust to changes in $w$ when $w$ is within the range of 0 to 1.

The results indicate that effective reward function design in RFT requires incorporating reasonable constraints (*e.g.*, $\delta$ in the IC reward) to prevent against reward hacking.

Table 6: Ablation studies of $\delta$ and $w$ in reward function.

| Parameter Value | QVHighlights | |
|---|---|---|
| | R@1 (IoU=0.5) | R@1 (IoU=0.7) |
| *Ablation of $\delta$, set $w = 0.5$* | | |
| $\delta = 0.2$ | 24.4 | 10.0 |
| $\delta = 0.5$ | 34.3 | 19.0 |
| $\delta = 0.8$ | 33.1 | 16.8 |
| *Ablation of $w$, set $\delta = 0.5$* | | |
| $w = 0.2$ | 34.8 | 18.9 |
| $w = 0.5$ | 34.3 | 19.0 |
| $w = 0.8$ | 34.8 | 20.7 |

## A.2  IMPACT OF NUMBER OF TEST FRAMES

As discussed in the main text, we have examined the effect of the number of sampled frames during training. Here, we present the impact of the number of test frames on performance in Table 7. Specifically, we sample 128 frames during training and evaluate using 32, 64, 128 and 192 frames on both long and short video temporal grounding benchmarks. Based on the experimental results, we draw the following conclusions.

On the long video temporal grounding benchmark, performance improves as the number of test frames increases. This is because long videos contain more complex semantic information and a more dispersed distribution of events. Fewer frames result in larger temporal gaps, potentially leading to the loss of critical contextual information, which consequently degrades the model's performance.

On the short video temporal grounding benchmark, an excessively high number of test frames negatively impacts performance. A moderate number of test frames yields optimal results. This can be attributed to the high frame similarity in short videos. An excess of frames introduces redundant information, which interferes with the model's decision-making process.

Table 7: Impact of number of test frames.

| Test Frame Number | Charades-STA (Short) | | QVHighlights (Long) | |
|---|---|---|---|---|
| | R@0.5 | R@0.7 | R@0.5 | R@0.7 |
| 32 | 34.2 | 14.3 | 15.6 | 5.9 |
| 64 | **36.6** | **14.8** | 16.5 | 7.9 |
| 128 | 32.2 | 13.0 | 34.3 | 19.0 |
| 192 | 22.0 | 8.7 | **39.0** | **19.6** |

### A.3 IMPACT OF MODEL SIZE

We conduct an ablation study of model size, with the results shown in Table 8. The 7B model demonstrates superior long video temporal grounding performance compared to the 3B model. This indicates that a larger parameter size effectively enhances the model's reasoning capability.

Table 8: Impact of model size.

| Model Size | QVHighlights | |
|---|---|---|
| | R@1 (IoU=0.5) | R@1 (IoU=0.7) |
| 3 B | 15.5 | 8.0 |
| 7 B | 34.3 | 19.0 |

## B DATASET FILTERING DETAILS

Data quality is crucial for the stability and performance of reinforcement fine-tuning. Inspired by FAST-GRPO (Xiao et al., 2025), we also adopt a similar data filtering strategy. The filtering process consists of the following two stages.

**Stage 1: Video length filtering.** We collect videos from the InternVid (Wang et al., 2023) dataset as source data. All videos longer than 60 seconds are removed, ensuring that the training dataset consists entirely of short videos. This stage results in 71K samples.

**Stage 2: Difficulty-based filtering.** The core principle of this stage is that moderately difficult samples are the most beneficial for reinforcement fine-tuning, as excessively simple or difficult samples result in sparse optimization signals. We utilize the base model Qwen2.5-VL-7B-Instruct (Bai et al., 2025) to generate 6 predictions for each sample and compute the IoU with the ground truth. The average IoU is then used as a metric for data selection. The distribution of IoU values is presented in the Figure 7. Among them, samples with an IoU between 0 and 0.1 account for the majority, which are identified as difficult cases during the filtering process. In contrast, samples with an IOU greater than 0.5 are less frequent and are considered easy cases. We select samples with IoU between 0.1 and 0.5, resulting in a final training dataset of 16K samples for reinforcement fine-tuning.

## C MORE IMPLEMENTATION DETAILS

### C.1 HYPER-PARAMETER LIST

We provide the detailed training setting and hyper-parameter values in Table 9 to facilitate future reproducibility.

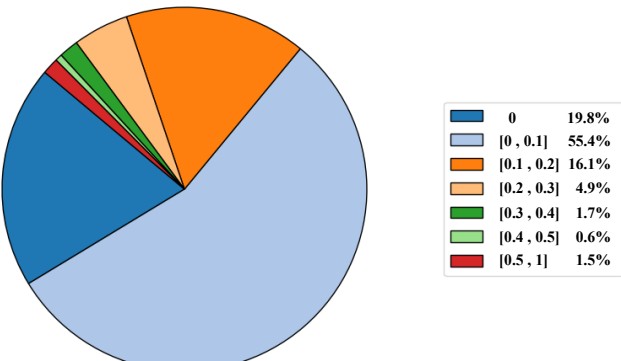

Figure 7: IoU distribution during difficulty-based filtering.

Table 9: Training setting and hyper-parameter list.

| Hyper-parameter | Value |
|---|---|
| Base model | Qwen2.5-VL-7B-Insruct |
| Training Epochs | 1 |
| Batch size per device | 1 |
| Deepspeed setup | Zero3 Offload |
| Learning rate | 1e-06 |
| Optimizer | AdamW |
| Optimizer momentum | 0.9 0.99 |
| Weight decay | 0.0 |
| Frame number | 128 |
| Frame resolution | $224 \times 224$ |
| Max prompt length | 8912 |
| Max completion length | 32768 |
| Temperature | 0.9 |
| Rollout number | 6 |
| KL coefficient | 0.04 |
| IC reward $\delta$ | 0.5 |
| IC reward $w$ | 0.5 |

## C.2 PROMPT DESIGN

In this paper, we adopt relative frame order to replace absolute timestamps within the reinforcement fine-tuning framework. In the prompt, the model is required to predict the start and end frame indices of the target event instead of precise numerical values. The prompt based on relative frame order is shown in Figure 8(a), while Figure 8(b) demonstrates prompt using absolute time.

## C.3 BENCHMARK DATASETS

In the main text, we briefly introduce four long-video temporal grounding benchmarks. A more detailed description is provided in this section.

- **QVHighlights.** This open-domain dataset (Lei et al., 2021) consists of 1.5K videos, each lasting 150 seconds. It provides 1.5K annotated samples in total, with an average event duration of 32 seconds. The videos mainly consist of news reports and vlogs.

---

### *Prompt of Relative Frame Order*

""" **For a video, we sample 128 frames.** In order to accurately pinpoint the event "[EVENT]" in the video, please determine **the start frame number** (This event starts from) and **the end frame number** (This event ends). The start frame number and the end frame number **are integers between 1 and 128.**

Output your thinking process in the <think> </think> tags. You should analyze the association between the visual information **of different frames and events** to determine between **which two frames the event occurred**.

Then, provide the start frame number and end frame number in the format of **"start frame to end frame"** in the <answer> </answer> tags (using integers from 1 to 128). **For example: "2 to 8".** """

(a) Prompt template with relative temporal representation

---

### *Prompt of Absolute Timestamps*

""" To accurately pinpoint the event "[EVENT]" in the video, determine the **precise time period of the event**.

Output your thought process within the <think> </think> tags, including analysis with either **specific timestamps (xx.xx) or time ranges (xx.xx to xx.xx).**

Then, provide the start and end times (in seconds, precise to two decimal places) in the format **"start time to end time"** within the <answer> </answer> tags. **For example: "12.54 to 17.83".** """

(b) Prompt template with absolute temporal representation

Figure 8: Prompt designed for different temporal representations.

- **ActivityNet-Long.** We select videos longer than 180 seconds from the ActivityNet Captions (Caba Heilbron et al., 2015) test set, resulting in a total of 3.5K samples, with an average video length of 209 seconds. This dataset can be regarded as a more challenging subset filtered from ActivityNet Captions.

- **HiREST.** HiREST (Zala et al., 2023) primarily consists of videos from cooking and assembly domains, where events exhibit strong temporal dependencies. The test set includes 77 videos with an average duration of 261 seconds. Events are annotated at a fine temporal granularity, with an average length of 19 seconds, making the temporal grounding task particularly challenging.

- **TACoS.** TACoS (Regneri et al., 2013) focuses on the cooking domain and includes both egocentric and exocentric views. It comprises 25 videos with an average duration of 368 seconds, resulting in 4K samples. The average duration of events is 32 seconds.

### C.4 COMPARISON OF SETTINGS

In this paper, we introduce a new setting called **Short-to-Long**. This section provides a detailed comparison with previous setting.

- **Previous Setting.** Previous methods employ multi-task supervised learning fine-tuning for training. As shown in Table 10, the training set of TimeChat (Ren et al., 2024) includes 6 tasks and 11 datasets, with a total of 125K time-related samples. Based on this, VTG-LLM (Guo et al., 2025b) filters and re-annotates this dataset, resulting in a final training set of 120K samples. TimeSuite (Zeng et al., 2024) contains 349K time-related data samples and 49K general video-dialogue pairs. However, they only evaluate on the short-video temporal grounding benchmark, Charades-STA (Gao et al., 2017). In this setting, the test samples are significantly simpler than the training samples.
- **Short-to-Long Setting.** Our proposed setting removes other time-related tasks and focuses solely on video temporal grounding. The training set consists of short videos under 60 seconds. After filtering, our training set contains only 16K samples, with an average duration of 38 seconds. In contrast, the test datasets are composed of long videos, with an average length ranging from 150 to 368 seconds. In this setting, the test samples are more challenging than the training samples.

Table 10: Multi-task training of TimeChat (Ren et al., 2024).

| Task | Dataset |
|---|---|
| Video Temporal Grounding | DideMo, QueryD, HiREST |
| Step Localization | COIN, HiREST |
| Dense Video Captioning | ActivityNet Captions, ViTT |
| Video Summarization | TVSum, SumMe |
| Video Highlight Detection | QVHighlights |
| Speech Generation | YT-Temporal |

## D  LIMITATIONS AND FUTURE WORK

### D.1  LIMITATIONS

Although VTG-Reasoner enhances the model's reasoning ability for long video temporal grounding, there are still limitations that warrant further exploration.

First, our method performs less effectively when the number of sampled frames is limited. That is because a smaller number of sampled frames results in larger temporal gaps between frames, leading to coarse-grained grounding results. We hope that a well-designed token compression mechanism can help alleviate this issue. Second, although we introduce the Short-to-Long setting, our model still struggles to achieve satisfactory temporal grounding performance on videos of hour-level duration. For extremely long video domains, more specialized designs, such as iterative grounding strategies, remain to be explored.

### D.2  FUTURE WORK

This paper highlights the potential of RFT for long video temporal grounding. We propose two promising directions for future exploration.

- Scaling VTG-Reasoner to accommodate longer videos such as hour-long videos.
- Investigating the effectiveness of RFT in general long-video understanding tasks.

We hope that these insights could inspire the community to develop more robust, generalizable, and powerful video understanding models.

## E  MORE CASE STUDIES

In this section, we provide additional case studies. Figures 9, Figure 10, and Figure 11 show case studies from the HiREST, QVHighlights, and TACoS datasets, respectively. Visualization results

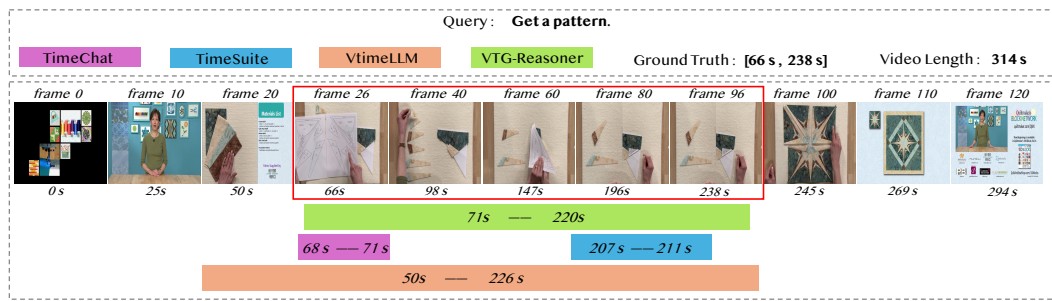

*Reasoning of VTG-Reasoner :* *<think>* The event "Get a pattern" likely refers to the moment when the person begins to lay out or arrange the fabric pieces according to the pattern. This typically **happens after the materials list has been shown** and the person has gathered all necessary pieces. Based on the sequence of actions, it seems that the person starts arranging the fabric pieces around the central star pattern **after showing the pattern sheet**. **From the beginning of the video to frame 30**, a person introduces and presents the materials list. **After that**, the pattern is made. From frame 90 to the end of the video, the patterns are arranged. Therefore, the event "Get a pattern" **would occur between the introduction of the pattern sheet and the arrangement of the fabric pieces**. That is, from frame 30 to frame 90.*</think>*

Figure 9: Case study from HiREST dataset.

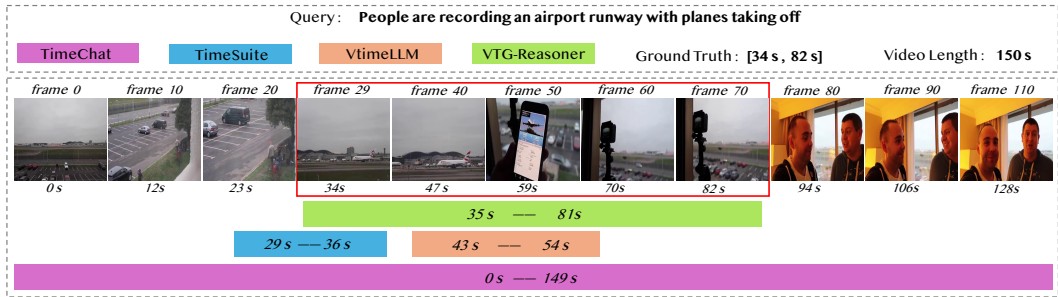

*Reasoning of VTG-Reasoner :* *<think>* The event "People are recording an airport runway with planes taking off" **occurs when the camera focuses on** the airport runway and captures planes taking off. **This happens around frame 30**, where the camera shows a plane taking off from the runway. **The event continues until frame 70**, where **the camera shifts** focus to two men in a room, **indicating the end of the event**. *</think>*

Figure 10: Case study from QVHighlights dataset.

demonstrate that other methods struggle to achieve precise temporal grounding. For example, the predicted intervals of TimeSuite are quite small, whereas VtimeLLM tends to predict larger intervals. In contrast, VTG-Reasoner effectively leverages the video context to produce a logical reasoning process, which results in relatively accurate temporal grounding predictions for long videos.

Additionally, we present a failure case in Figure 12. In this instance, VTG-Reasoner fails to accurately analyze the relationship between the event and the video, leading to an overly large predicted time interval. This exemplifies the phenomenon of greedy predictions discussed in the main text. Notably, we observe a significant reduction in such failure cases following the incorporation of the IC reward.

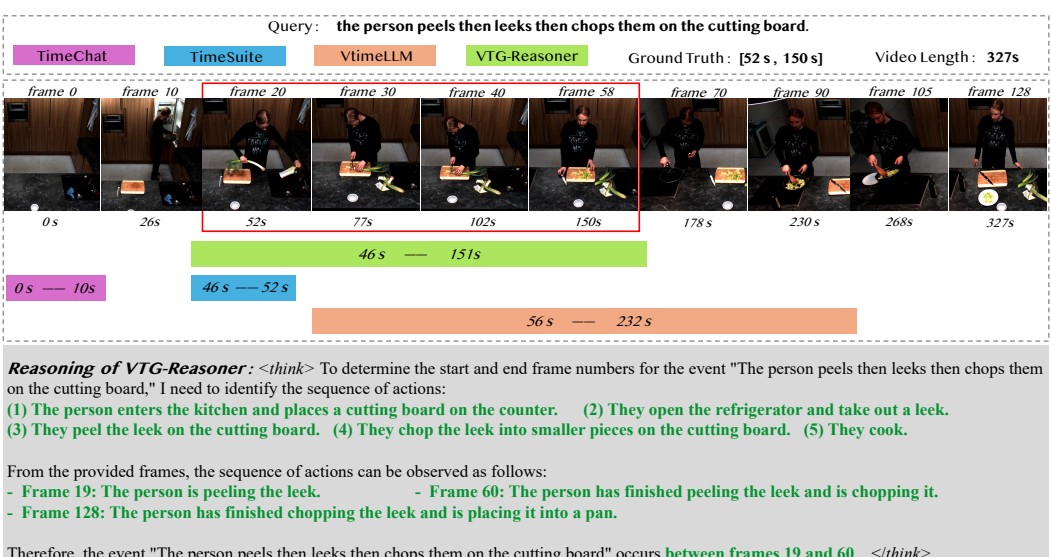

Figure 11: Case study from TACoS dataset.

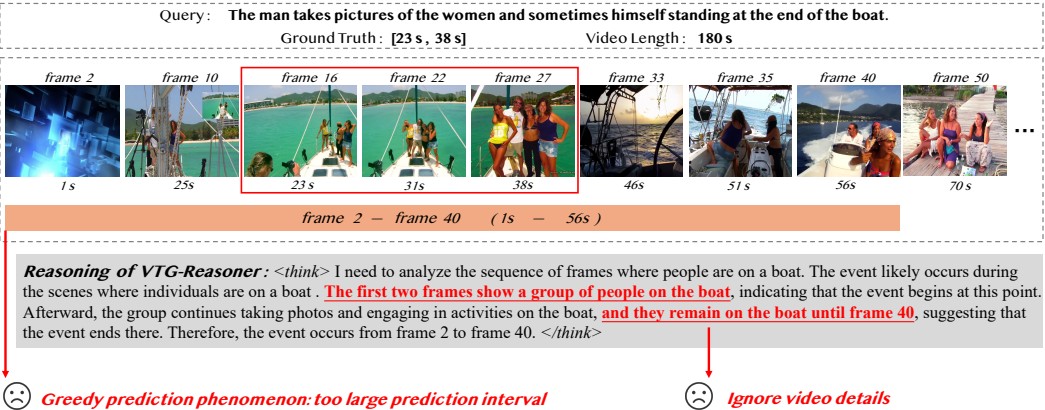

Figure 12: A failure case.

