# OpenReview forum: "VTG-Reasoner: Long Video Temporal Grounding via Reinforcement Fine-Tuning"
_ICLR.cc/2026/Conference — ICLR 2026 Conference Withdrawn Submission_

### Official Review · Reviewer_1kd6 · 2025-10-14

**Soundness:** 2
**Presentation:** 3
**Contribution:** 2
**Rating:** 4
**Confidence:** 4

**Summary:**

This paper introduces VTG-Reasoner, a Reinforcement Fine-tuning (RFT) framework aimed at improving long-video temporal grounding. The authors posit that traditional Supervised Fine-tuning (SFT) methods tend to overfit the training data, leading to poor generalization. To address this, the proposed method utilizes the GRPO algorithm to optimize the model with a reward function combining IoU and a novel Intersection Compactness (IC) metric. Key aspects of the methodology include the use of relative frame order instead of absolute timestamps and the introduction of a challenging "Short-to-Long" evaluation protocol. Experimental results demonstrate that VTG-Reasoner achieves strong performance on four long-video temporal grounding benchmarks.

**Strengths:**

* The paper correctly identifies a key limitation of SFT-based methods for this task. Using a cross-entropy loss on generated tokens struggles to measure the actual temporal distance between a close prediction and the ground truth. In contrast, the RFT approach with an IoU-based reward provides a much more direct and suitable optimization signal for evaluating the accuracy of predicted timestamps.

* The design of the Intersection Compactness (IC) reward is well-motivated. It effectively addresses the common failure mode where models predict overly large intervals to maximize the IoU score. As shown in Table 4, this component proves to be effective in encouraging more precise localizations.

**Weaknesses:**

* The central idea of using GRPO-based reinforcement learning with an IoU reward for video temporal grounding has been explored in several prior works [1-5]. The paper does not sufficiently differentiate its core RFT approach from these existing methods or clearly articulate its unique contributions in this context.

* While effective, the use of relative frame order is a common and almost consensus practice in recent video grounding models. Therefore, presenting it as a key contribution of this work might be an overstatement.

* A minor question, the paper overlooks several recently published and highly relevant papers in video temporal grounding [6-9].


[1] Video-R1: Reinforcing Video Reasoning in MLLMs

[2] Time-R1: Post-Training Large Vision Language Model for Temporal Video Grounding

[3] Reinforcement Learning Tuning for VideoLLMs: Reward Design and Data Efficiency

[4] videoChat-R1: Enhancing Spatio-Temporal Perception via Reinforcement Fine-Tuning

[5] MUSEG: Reinforcing Video Temporal Understanding via Timestamp-Aware Multi-Segment Grounding

[6] Universal Video Temporal Grounding with Generative Multi-modal Large Language Models

[7] TRACE: Temporal Grounding Video LLM via Causal Event Modeling

[8] Grounded-VideoLLM: Sharpening Fine-grained Temporal Grounding in Video Large Language Models

[9] LLaVA-ST: A Multimodal Large Language Model for Fine-Grained Spatial-Temporal Understanding

**Questions:**

Please see Weaknesses

---

### Official Review · Reviewer_82Dp · 2025-10-30

**Soundness:** 3
**Presentation:** 3
**Contribution:** 3
**Rating:** 6
**Confidence:** 3

**Summary:**

This paper proposes VTG-Reasoner, a reinforcement fine-tuning framework that improves the reasoning ability of MLLMs for long-video temporal grounding. Unlike standard SFT approaches that directly regress timestamps, the authors design an RFT pipeline based on GRPO, allowing the model to explore multiple reasoning chains and candidate intervals before updating with explicit rewards.
Two reward functions are introduced to balance accuracy and compactness of predicted temporal segments. Furthermore, they replace absolute timestamps with relative frame indices, providing a unified temporal representation across videos of varying duration. Under a proposed Short-to-Long training setup, VTG-Reasoner shows consistent gains on four long-video grounding benchmarks.

**Strengths:**

- The paper’s key contribution is applying RFT to temporal grounding in long videos, a setting not well explored in previous MLLM works. The use of relative frame order as temporal representation is a simple but elegant design that alleviates distribution shift between short and long videos. The Short-to-Long benchmark setting provides a meaningful evaluation protocol emphasizing data efficiency and generalization.

- The paper builds on the well-established GRPO algorithm and adapts it carefully to multimodal grounding tasks. Experiments cover four public benchmarks with clear metrics and comparisons.

**Weaknesses:**

- It remains unclear how the adopted RFT changes the model’s internal reasoning or attention patterns. There is no quantitative measure of reasoning trace quality provided.

- The authors briefly note performance degradation with fewer sampled frames, but more reflection on temporal ambiguity and multi-event queries would be helpful.

- The “Short-to-Long” setting is interesting but somewhat self-defined; comparisons are made under different training data scales, which may not be fully fair. Further justifications regarding this are expected.

- It is recommended to include some recent video grounding works from ICCV 2025 to better position this paper against the current state of the art, such as Vid-Group: Temporal Video Grounding Pretraining from Unlabeled Videos in the Wild and OVG-HQ: Online Video Grounding with Hybrid-modal Queries.

**Questions:**

NA

---

### Official Review · Reviewer_44d5 · 2025-10-31

**Soundness:** 3
**Presentation:** 3
**Contribution:** 2
**Rating:** 2
**Confidence:** 4

**Summary:**

The paper presents VTG-Reasoner, a GRPO-based reinforcement fine-tuning framework for long-video temporal grounding. Two main ideas are emphasized: (1) a reward design that combines standard IoU with a newly proposed Intersection Compactness term to discourage overly long predictions; and (2) supervising relative frame indices instead of absolute timestamps. The authors further adopt a Short-to-Long evaluation protocol, which trains on short clips and tests on long videos, reporting gains over several grounding baselines.

**Strengths:**

1. **Evaluation protocol**: Training solely on short videos and testing on long videos is a clear, relevant setup to verify length generalization.
2. **Method clarity and ablations**: The approach is easy to follow, and the ablations cover key designs (reward components, supervision target, etc).
3. The writing is generally clear and the paper is straightforward to follow.

**Weaknesses:**

1. **Limited technical novelty**: Beyond the IC reward, the approach is largely incremental relative to recent RL-based grounding systems (e.g., VideoChat-R1).
2. **Missing comparisons with existing RL methods**: Several existing RL-based temporal grounding methods (e.g., VideoChat-R1, Time-R1, DeepVideo-R1, Thinking with Videos, etc.) are not compared. On datasets such as ActivityNet and Charades-STA, the reported numbers for VTG-Reasoner appear weaker than some of these methods, but the paper lacks discussion or diagnostic analysis explaining why.
3. Sensitivity of relative-index supervision: Supervising relative frame positions can hurt robustness when the test-time frame number sampling differs from training, which is very common in video VLM practice. The ablations in Table 7 suggest performance depends on a careful choice of frames at inference, which complicates the claimed generalization benefits.

**Questions:**

Please see the weaknesses 2 and 3. Particularly, please compare to existing RL-based grounding methods, or explain clearly why such comparisons are impractical.

---

### Official Review · Reviewer_bwjP · 2025-11-08

**Soundness:** 3
**Presentation:** 2
**Contribution:** 2
**Rating:** 4
**Confidence:** 4

**Summary:**

This paper proposes VTG-Reasoner, a reinforcement fine-tuning (RFT) framework for long video temporal grounding(VTG). The goal is to locate the start and end timestamps of an event described by a natural language query within a long video. While existing methods (e.g., TimeChat, TimeSuite, HawkEye) mainly use supervised fine-tuning (SFT) on short videos, this work identifies that such models generalize poorly to longer videos. VTG-Reasoner aims to enhance reasoning ability with the following components: 1) Adopting Group Relative Policy Optimization (GRPO) for multi-round exploration and trial-and-error reasoning; 2) introducing a novel reward functions; 3) unifying temporal representations across varying lengths videos using relative frame numbers instead of absolute timestamps; and 4) proposing a Short-to-Long setup, where models are trained only on short (<60s) videos and evaluated on long (>150s) videos. Experiments show VTG-Reasoner significantly outperforms SFT-based baselines on four long video benchmarks, including QVHighlights, ActivityNet-Long, HiREST, and TACoS with only 16K training samples, while maintaining (or improving) performance on short-video datasets and VQA benchmarks (MVBench, VideoMME).

**Strengths:**

**[S1]** This paper is well-motivated, addressing a clear gap in long video grounding.

**[S2]** Ablations are comprehensive, demonstrating the effectiveness of each component in the proposed method. Especially, compactness reward is intuitive and shown effective in ablation.

**[S3]** The proposed method is data efficient, achieving better performance using fewer videos than baselines.

**Weaknesses:**

**[W1]** Fairness of comparisons
- The proposed method and baselines employ different models as backbones (e.g. ViT-G/14 and LLaMA-2 of TimeChat and Qwen2.5-VL-7B-Instruct of this work). The comparison may be unfair from this perspective. Presenting and comparing the zero-shot performance of base models may provide a guideline for fair comparison.
- Some baselines keep pre-trained LLMs frozen, while the proposed method seems to fine-tune the whole parameters of Qwen2.5-VL-7B-Instruct. Although the proposed method requires fewer data to achieve the performance, the number of fine-tuned parameters is also a significant part in this work.
- The Qwen2.5-VL already has the capability in ultra-long video understanding and fine-grained video grounding, as presented in the technical report of the model. Applying the proposed method to various base models would be better to highlight the effectiveness and contributions.

**[W2]** RFT
- There is no comparison to PPO-based RFT or analysis of why GRPO was chosen.
- There is no discussion of reward variance or stability, which is known to affect RFT.
- There is no mention of reward normalization or baseline subtraction besides group-wise mean/std; convergence curves are missing.

**Minor issues**
- Typo: L321 privious → previous

**Questions:**

**[Q1]** Relative frame order for timestamps. Typically, video temporal grounding models have been trained with normalized timestamps for efficient batch training. In the reviewer’s understanding, the relative frame order-based timestamps can be regarded as such normalized timestamps. What is the major difference between them, and what can be a new insight for the proposed approach?

**[Q2]** How stable is training under GRPO? Is there any instability or divergence observed (e.g., reward oscillation)?

**[Q3]** L31 “Unlike semantic question answering, ~”: There are several benchmarks that require fine-grained modeling for accurate answer prediction. This phrase can be misunderstood.

---

### Note · Authors · 2025-11-13

I have read and agree with the venue's withdrawal policy on behalf of myself and my co-authors.